# Towards a Non-Contact Method for Identifying Stress Using Remote Photoplethysmography in Academic Environments

**DOI:** 10.3390/s22103780

**Published:** 2022-05-16

**Authors:** Hector Manuel Morales-Fajardo, Jorge Rodríguez-Arce, Alejandro Gutiérrez-Cedeño, José Caballero Viñas, José Javier Reyes-Lagos, Eric Alonso Abarca-Castro, Claudia Ivette Ledesma-Ramírez, Adriana H. Vilchis-González

**Affiliations:** 1School of Engineering, Universidad Autónoma del Estado de México, Toluca de Lerdo 50100, Mexico; hmoralesf001@alumno.uaemex.mx (H.M.M.-F.); jcaballerov@uaemex.mx (J.C.V.); avilchisg@uaemex.mx (A.H.V.-G.); 2School of Medicine, Universidad Autónoma del Estado de México, Toluca de Lerdo 50180, Mexico; jjreyesl@uaemex.mx (J.J.R.-L.); ciledesmar@uaemex.mx (C.I.L.-R.); 3School of Behavioral Sciences, Universidad Autónoma del Estado de México, Toluca de Lerdo 50010, Mexico; agutierrezc@uaemex.mx; 4División de Ciencias Biológicas y de la Salud (Health and Biological Sciences Division), Universidad Autónoma Metropolitana, Lerma de Villada 52006, Mexico; e.abarca@correo.ler.uam.mx

**Keywords:** academic stress, remote photoplethysmography, classifiers, stress recognition, anxiety

## Abstract

Stress has become a common condition and is one of the chief causes of university course disenrollment. Most of the studies and tests on academic stress have been conducted in research labs or controlled environments, but these tests can not be extended to a real academic environment due to their complexity. Academic stress presents different associated symptoms, anxiety being one of the most common. This study focuses on anxiety derived from academic activities. This study aims to validate the following hypothesis: by using a non-contact method based on the use of remote photoplethysmography (rPPG), it is possible to identify academic stress levels with an accuracy greater than or equal to that of previous works which used contact methods. rPPG signals from 56 first-year engineering undergraduate students were recorded during an experimental task. The results show that the rPPG signals combined with students’ demographic data and psychological scales (the State–Trait Anxiety Inventory) improve the accuracy of different classification methods. Moreover, the results demonstrate that the proposed method provides 96% accuracy by using K-nearest neighbors, J48, and random forest classifiers. The performance metrics show better or equal accuracy compared to other contact methods. In general, this study demonstrates that it is possible to implement a low-cost method for identifying academic stress levels in educational environments.

## 1. Introduction

Stress is an extremely common health condition. Signs of stress are observed in people of all ages as a result of daily activities. Undergraduate students typically experience stress as a result of assignments, class activities, and/or exams. Academic stress refers to the stress experienced by students in academic environments. Students typically lack strategies for coping with stress and, consequently, may experience adverse psychological effects.

Stress identification is not new and has been researched from different perspectives. Nevertheless, the definition of stress is unclear. In the clinical literature, there is a lack of agreement regarding the definition of stress, and its different definitions are often complex [1]. The term stress is often confused with anxiety. These terms are sometimes used interchangeably [2]. According to Selye [3], stress refers to unpleasant feelings aroused by emotional, physical, or mental challenges that subjects encounter in their lives. Stress is typically caused by an external trigger [4,5]. On the other hand, anxiety is defined as excessive and persistent worries that do not subside even when the stressor is no longer present [6].

Identifying stress without the help of an expert is challenging. A stress response leads to psychological and physiological symptoms, which are produced when stress stimulates the sympathetic nervous system (SNS). The measurement of changes in physiological and/or psychological parameters can be used to identify subjects with chronic stress, assuming that these changes cause SNS reactions triggered by chronic stress [7]. These changes are usually evaluated by way of self-evaluation questionnaires or inventories. Nevertheless, the link between stress and responses to different situations is not straightforward. Stressful evaluative situations commonly elicit feelings of anxiety [8]. Moreover, previous studies [9,10] describe state anxiety as an emotional response to a subject’s perception of a stressful experience. In this way, stress can be measured by observing anxiety symptoms while subjects perform specific tasks. Psychological inventories are used for this purpose. The results of these inventories give a suitable and accurate measure of anxiety.

Recent studies suggest that academic stress is related to low academic performance and student disenrollment [4]. Kitsantas and Cols [11] claim that up to 75% of undergraduate students experience stress during their first year of university. In addition, Ros-González [12] shows that stress causes up to 50% of first-year university dropouts. Stress causes disenrollment from public universities, which is a considerable cost to society.

Hassard et al. [13] developed a systematic approach for identifying the costs to society of work-related stress. Between 70% and 90% of cases of work-related stress lead to falls in productivity. Between 10% and 30% of cases of work-related stress lead to medical expenses. The early detection of academic stress is not only crucial for avoiding student crises but also for avoiding economic problems.

Research on academic stress discusses the relationship between stress and anxiety. Anxiety can be triggered by academic stress when students face demanding school situations (stressors), for example examinations. De Witte et al. [14] claim that stress-related emotional states lead to anxiety. Thus, students must use all their coping strategies to adapt or face state anxiety. In this process, the associated stress symptoms as well as adaptive, behavioral, emotional, and anxious features are present. Anxiety is a crucial component of the academic stress observed in anxious behaviors and symptoms of anxiety or distress. The effects could have an impact on the balance of the students’ psychological, cognitive, social, and affective states; even in the most critical cases. Maturana and Vargas [1] define anxiety states as part of the clinical consequences of academic stress in students with repercussions on their quality of life and academic performance.

Stress detection from body response signals is not new, either. However, current systems and techniques are primarily low cost and intrusive, or they are high cost and non-intrusive [15]. Different studies have shown that stress can be summarized as a set of observable physiological signs such as sweating [16,17], increased heart rate [18], breathing [19], facial gestures [20] or quick eye blinking [20]. The correlations between stress levels and heart rate, oxygen saturation, or galvanic skin response have been researched, and acceptable results have been obtained [21,22]. However, these tests have not been applied explicitly to academic environments. Academic stress, as defined by Putwain [4], consists of all related activities and behaviors that occur at school or are associated with any school-related task. Most of the studies on academic stress and anxiety have been conducted in research labs or controlled environments. Such environments might lead to stress and/or anxiety states. Alberti et al. [23] ran a series of detailed tests to measure stress in a multimodal environment. They obtained impressive results by using different types of sensors and techniques. Unfortunately, these tests can not be extended to a real academic environment due to their complexity. In general, academic stress detection systems are too complex, expensive and/or impractical for use in schools and universities [24,25].

This work identifies academic stress (stress derived from academic activities) as the stressor (for example, excessive responsibilities, high workload, or tasks inside and outside the academic environment, teacher evaluations, among others). A non-contact method for identifying states of academic stress and its associated anxiety features using remote photoplethysmography in combination with students’ demographic data is proposed.

Anxiety is one of the emotions resulting from stress and one of the clinical consequences of academic stress [26]. Different inventories and scales have been developed to measure anxiety levels [27,28]. One frequently used instrument to measure anxiety levels in the field of psychology is the State–Trait Anxiety Inventory-STAI [29,30]. It consists of two self-evaluation scales that are used to measure Trait Anxiety (A-trait), or SXR scale, and State Anxiety (A-state), or SXE scale. The advantages of the STAI assessment are the short time required to answer both scales and the easy interpretation of results [30]. It has been demonstrated in research that subjects obtain a high anxiety score when in an anxiety-inducing situation [31].

In the case of this research, we have focused on the anxiety symptoms caused by academic stress as one of the clinical consequences. We therefore regard state anxiety as an academic stress-related outcome. The STAI assessment has been widely used as ground truth to evaluate subjects with anxiety [7,32,33]. Consequently, the STAI is a suitable tool for the assessment of anxiety characteristics in academic environments due to its clinical validity. It is relatively quick to conduct STAI assessments because students rarely find difficulty in answering them.

A-trait anxiety is based on twenty items which ask the respondent how they feel in general. A-state anxiety also has twenty statements, but directions require individuals to provide answers based on how they feel right now, that is, at the moment of answering the questionnaire, or immediately following a particular situation, for example during a class or exam. The minimum and maximum scores for both SXR and SXE scales are 20 and 80, respectively. Values of less than 36.99 for males or 37.24 for females are considered normal. Values higher than 43.01 for males and 43.69 for females are considered high. The subject responds to each of the questions by assessing themselves on a 4-point Likert scale. The SXE scale can be used to determine the anxiety level produced by any given experimental task, such as an anxiety-inducing situation. According to the STAI user manual [29], a low, medium or high anxiety level will be observed in subjects in specific anxiety situations, for example, in the case of this study when students perform academic activities or take an exam.

Certain physiological responses correlate to stress. There are several state-of-the-art techniques which identify stress states using physiological signals. Heart rate has been widely used as an indicator of stress-induced anxiety. Another technique is photoplethysmography (PPG) [34], which is a simple and low-cost optical technique that can be used to detect blood volume changes in microvascular beds. This technique is accurate, but it is not suitable for use in academic environments. In a previous study, the authors [35] propose a system for measuring academic stress using an experimental setup to replicate academic activities. Wired sensors are used during emulated tests in different phases: no stress, low stress, and high stress. Nevertheless, the main limitation of this platform is that it is not feasible to use in class due to the set-up time required. Furthermore, the movements and interactions of the students are limited by a sensor being placed on one of the hands.

Non-contact techniques are the most recent advances in the development of innovative tools for obtaining physiological data. Some techniques are based on pattern recognition [36], using video-based image recognition. The use of low-cost cameras or built-in cameras on mobile phones or laptops are practical ways of collecting data. Video-based stress detection systems use a bio-signal that can be allocated in any of the three ways shown in Table 1.

This study proposes using a type C system based on remote photoplethysmography (rPPG) [42,45]. rPPG is a non-contact video-based technique that measures changes in blood volume by analyzing pixel intensity changes from an image of the subject’s skin to obtain the heart rate (HR) value. For video-based systems to detect heart rate, the quality of the video must be high, the lighting must be good, and a suitable face-detection algorithm must be used. Such an algorithm provides the necessary data for a PPG signal. This can be used to calculate a set of characteristics for classification. Then, with the data obtained using rPPG, the system calculates a pulse rate value that can be correlated with anxiety reactions based on the results of the STAI assessment.

This research aims to validate the following hypothesis: by using a non-contact method (video recordings and rPPG), it is possible to identify academic stress levels with an accuracy greater than or equal to those reported in previous works, which used contact techniques.

## 2. Materials and Methods

### 2.1. Participants

A total of 56 first-year undergraduate students were observed, 34 students were male and 22 students were female (between 18 and 24 years of age). All of them were informed of the goal of the experiment, the procedures, data collection, including video recordings of their face, personal data, and demographic information. All students provided their consent voluntarily. They were informed that if they did not want to participate in any of the sessions for whatever reason, they could leave the experiment at any time.

### 2.2. Experimental Protocol

The methodology applied in this study consists of 4 phases, and these are shown in Figure 1:ARun the experimental academic stress protocol.BComplete the STAI assessment and student’s profile. Record the video during the academic tasks.CCalculate the HR values from video recording, using rPPG.DTrain and validate a classifier to identify stress using HR values and students’ profile (demographic data).

The first step in the methodology was to define and run the experimental protocol to measure stress. In this case, a STAI assessment was used while students were doing a specific academic activity. According to Putwain [4], this can be a class, a homework assignment, or an exam. The objective was to measure anxiety due to academic stress reactions during any of these activities in a non-contact way; efforts to reduce any stressors which were not related to the task were made. An experimental protocol was defined, and students’ faces were recorded during different academic tasks. In contrast to the procedure proposed by Rodriguez-Arce et al. [35], here, STAI scales were completed during a baseline activity of normal to mid anxiety levels to obtain trait anxiety (SXR) levels. In addition, in order to obtain SXE levels during mid to high anxiety academic activities, further testing was conducted during and after academic activities.

The STAI scale, validated for the Mexican population (Cronbach’s alpha = 0.87) and developed by Spielberger-Diaz-Guerrero [29], mentions that state scales can be obtained by adapting instructions to influence the behavior of the subject. The proposed protocol was designed as follows:Students complete both the trait (SXR) and state (SXE) scales during the baseline test, in this case, during a math class.Students fill out the state (SXE) scale during an academic evaluation, in this case, a written math exam.

Recurrent testing provided data on anxiety trend (SXR) and state anxiety (SXE) in different potentially stressful academic scenarios. In addition, rPPG was used to obtain the heart rate from a video source. Consequently, when the STAI scales were complete, the students’ faces were also recorded.

### 2.3. Video Recordings in the Academic Environment

The goal of the research was to identify stress in an academic environment without causing additional stress due to the use of tools or sensors. The procedure of completing STAI scales and recording the videos was a computational challenge in terms of data storage and network bandwidth. The students’ faces were recorded using a low-cost webcam. The videos were recorded at 30 frames per second, 480 × 640 pixels. The data from the STAI scales were collected through a web-based platform.

### 2.4. Calculate HR Using Video Recordings of Students’ Faces

The principal phase of the experimental methodology was to measure a physiological signal from students doing an academic task. In order to achieve this goal, a low-cost camera was used. As long as the device could record at more than 20 frames per second (fps) and at least 720p resolution, it was suitable for this experiment. Examples of such devices are built-in laptop webcams and mobile phone cameras. The minimum lighting requirements were standard classroom lighting either from a natural source or from any standard room lighting. Students looked directly at the camera.

The use of the DeHaan’s method to calculate the pulse rate from chrominance-based rPPG [42] provided heart-rate values (HR values) from students’ video recordings without the need for special lighting or body position setup. Subjects had to keep their faces in focus for the camera. In addition, they were asked to sit as still as possible. The technique consisted of obtaining a video recording from the camera to perform facial recognition and adjusting the Region of Interest (*ROI*) to students’ cheeks, from where a suitable pulse rate can be obtained [46]. Face recognition was done using the Haar feature-based cascade classifiers proposed by Viola and Jones [47], which provide pre-trained models of different facial objects and an algorithm to detect face features and discard what is not useful for face recognition.

Video recording was segmented in windows of 3.3 s, in which a pulse signal might be found [48]. DeHaan [42] recommends segments of at least 32 frames (less than 1.5 s in this study). Face recognition was performed within the interval in order to obtain ROIs in left and right cheeks. From those ROIs, RGB matrices were obtained, and the mean per ROI was calculated in order to normalize RGBs per ROI. In this way, the dominant pixel value from the interval was obtained.

Normalized RGB ROIs were transformed to XY, chrominance, luminance, and color space using DeHaan’s Equations (Equation 1) and (Equation 2), which provide skin color and non-white light compensation.
(1)$$X_{s}=3R_{n}-2G_{n}$$
(2)$$Y_{s}=1.5R_{n}+G_{n}-1.5B_{n}$$

$X_{s}$ and $Y_{s}$ signals were filtered using a FIR pass-band filter in frequencies between 0.67 Hz (40 bpm) and 3.67 Hz (220 bpm) as recommended by DeHaan.

The *X minus alpha Y* in Equation (Equation 3) was applied to the filtered $X_{f}$, $Y_{f}$ signals. Then, the $S_{f}$ signal, which contains the HR value of each ROI, was obtained.
(3)$$S_{f}=X_{f}-\alpha Y_{f}$$
with:(4)$$\alpha =\frac{\sigma (X_{f})}{\sigma (Y_{f})}$$
where σ refers to the standard deviation of the $X_{f}$ and $Y_{f}$ signals. The next step was to perform a convolution of the $S_{f}$ signal by a Hanning’s window in each interval to smoothen and filter the signals in the temporal space. Period and overlapped signals were summed after Hanning’s window was applied to obtain the interval’s HR signal.

Finally, *Fast Fourier Transform* was applied to the resultant *S_f_* signal in order to calculate the highest peak of the first harmonic, which contains the actual HR value of the actual window. Figure 2 illustrates the procedure.

The main advantage of using chroma-based rPPG was the response to non-light illumination, skin color masks, and movements of the face. Therefore, no particular setup in the classroom was needed.

In this study, an experimental platform was built to acquire the data, record the videos, process the raw data and videos, analyze the information, and provide an output for left and right cheeks corresponding to HR values. This initial prototype was developed using a Mac Book Pro^®^ 2015 with 2.6 GHz Intel Core i7, 16 GB DDR4, and 720p built-in camera. The implementation of chroma-based rPPG was done using Python 3.7, OpenCV 4.0, SciPy 1.3.1 and Matplotlib 3.1.1. From one minute of video, 18 HR values were obtained using 3.3 s frame intervals (the result of dividing 60 s into windows of 3.3 s). The goal of using the chroma-based rPPG was to calculate HR values while students completed an academic task.

### 2.5. Feature Extraction

The final step in the proposed methodology was to identify academic stress inferred from anxiety reactions through STAI scales in combination with the use of the students’ demographic data. As described in Section 2.3, demographic data were collected from students, and this information was used for classification: age (numeric); gender (Boolean, 0—male or 1—female); previous studies—private or public school (Boolean, 1—public, 0—private); student participation in sport (Boolean, 1—yes, 0—no); student participation in extra-curricular activities (Boolean, 1—yes, 0—no); family issues (Boolean, 1—yes, 0—no); failed classes (Boolean, 1—yes, 0—no).

The demographic data and the HR values of each student were mapped to the predictive label anxiety class, which was based on the results of the SXE scale. The cutoff scores of the SXE scale were set as a score of more than 43.01 for males and 43.69 for females. These were labeled as high academic stress (1); otherwise, the label was low academic stress (0). Figure 3 shows how these features were obtained and how the classifier predicted the label class for anxiety.

### 2.6. Classification of Stress Levels

The following classifiers were tested in this study:Support vector machine (SVM)—SVMs were initially proposed by Vapnik [49] for solving classification and regression problems. SVM is a supervised learning technique trained to classify different categories of data. One of the most important parameters is the kernel function (a way of using a linear classifier to solve a non-linear problem). The SVM model based on Pearson VII universal kernel (PUK) has been proposed in this study.K-nearest neighbors (KNN)—This is a supervised machine learning algorithm that can be used for regression and classification tasks. It was created as a result of the need to perform discriminant analysis when reliable parametric estimates of probability densities are unknown or difficult to determine [50]. The use of KNN is recommended when there is little or no prior knowledge of the data distribution. The most important parameter is the K value, which indicates the count of the nearest neighbors. In this study, K is set equal to 1.0.J48 decision tree—J48 is an open-source Java implementation of the C4.5 algorithm, which is used to generate decision trees. This algorithm was developed by Ross-Quinlan [51], and it is often referred to as a statistical classifier. The main parameters are the confidence factor and the minimum number of instances per leaf. In this study, the confidence factor and the value of the minimum number of instances are set to 0.25 and 2.00, respectively.Random forest (RF)—RFs are ensemble classifiers, which are used for classification and regression analysis of the data [52]. RF works by creating various decision trees in the training phase and output class labels those which have the majority vote [53]. There are three main parameters that must be configured, the number of trees to grow, seed, and tree depth. In this study, these parameters are set to 100, 1, and 0 (for unlimited), respectively.

These machine learning methods are common tools for classifying datasets when one has no prior knowledge of the data. They are broadly used and implemented in the Weka machine learning toolset [54]. For training and validation, k-fold cross-validation was used. Kuhn and Johnson [55] recommend the use of 10-fold cross-validation for small samples sizes due to the variance properties and desirable low bias of the performance estimate.

In order to evaluate the performance of each classifier, accuracy, sensitivity, and specificity were calculated using Equations (Equation 5)–(Equation 7). The Receiver Operating Characteristics curve (ROC) is a popular method for visualizing the tradeoffs between specificity and sensitivity in binary classifiers. The area under the curve (AUC) will be also calculated, and it is one way to summarize it in a single value. When AUC is near 1.0, this means that the classifier has a good measure of separability. A value near 0 means that the classifier has the worst measure of separability. When AUC is equal to 0.5, it means that the classifier has no class separation capacity.
(5)$$Accuracy=\frac{TP+TN}{TP+FP+TN+FN}$$
(6)$$Sensitivity=\frac{TP}{TP+FN}$$
(7)$$Specificity=\frac{TN}{TN+FP}$$

The Cohen’s Kappa coefficient [56] was calculated. This indicates the degree of agreement between two raters [57] in terms of accuracy: in this case, the agreement between the anxiety values provided by the STAI scales and the labels calculated by the classifier. Hence, it is a suitable statistical measure to accept or deny a classification accuracy. Table 2 shows the interpretation of Cohen’s Kappa values.

## 3. Case Study

In order to test the proposed methodology, an experiment was designed and run in which 56 students taking first-year classes in basic mathematics for engineering at a Mexican public university participated. At the beginning of the experiment, the students were informed of the purpose of the study, and written informed consent was obtained from each subject. Due to the COVID-19 pandemic, the ethics committee of our educational institution suspended its activities. However, all experimental procedures were conducted according to the ethical standards of the Declaration of Helsinki and following the relevant guidelines and regulations of our institution. Video recordings of students’ faces were obtained while they carried out an experimental task. In this case, the students were seated for a written math exam. At the end of the activity, they completed the STAI assessment. One of the contributions of this study is the use of rPPG in academic environments. This technique requires the use of videos of the participants. Our study involved the participation of subjects; however, no medical procedure was performed. Hence, this is not considered a clinical trial. Furthermore, the acquisition of videos was performed in an ethical manner, without violating the integrity and person of any student.

All experimentation took place in the facilities of the School of Engineering at the Universidad Autonóma del Estado de México. All experiments were run in standard classrooms; no special setup or conditions were needed.

A total of 56 first-year undergraduate students participated voluntarily in this study. The experiment was conducted in 2 sessions: (1) a class in which students’ demographic data and SXR/SXE scales were answered; and (2) an exam in which the SXE scale was completed. During both sessions, the students’ faces were recorded as described in Section 2.4. A total of 56 video files were recorded during the SXE/SXR class session, as well as 56 video files in the SXE exam session. In both cases, the average video recording length was 2 min.

### Methodology

The group of 56 students was all tested in a standard classroom. The group used the procedure illustrated in Figure 4:Academic stress protocol was defined to measure academic stress levels using anxiety as the metric during specific academic activities. The STAI was used to measure anxiety while videos were recorded during the academic stress test (activity).Two sessions were conducted: one baseline session to measure anxiety during a math class, and one stressful academic activity, in this case, a math exam. In both sessions, the professor was responsible for guiding the different experimental phases. He is one of the authors, and at any moment of the experiment, he provided enough information for the students to be aware of the experiment.During the baseline session (math class), students were asked to voluntarily step into the setup installed in the classroom and provide their profile information and complete the SXE/SXR scales. They were video recorded while doing this.Then, 45 min after the start of the written math exam, students were asked to voluntarily answer the SXE scale. They were video recorded while doing so. On average, students took 3 min to answer the 20 items. At the end of the activity, the students were informed that the exam would not affect their course grades.

The following data were acquired during this study:HR values from video recordings using rPPG.Students’ profile from the web platform.SXE scores from STAI assessment.

HR values were obtained as a 5-point moving average from HR signal values coming from the right and the left ROIs of the students’ faces using rPPG. Right and left HR values were averaged to obtain one single HR value vector for each student, and then, the 5-point moving average (referred to as intervals) was obtained to provide an HR value trend. This resultant HR value trend from the algorithm was referred to as HR-RAW data.

Figure 5 shows an example of HR-RAW data from one participant. There are two kinds of atypical values. In order to provide a clean, noise-free HR value vector, atypical HR values were removed. At the beginning of the vector, the first interval is always 0 (due to initial values) and must be removed.

Furthermore, at times, the participants moved. This caused their face to be momentarily out of focus. This caused some atypical values, which were removed. Although the participants were asked to face the camera, some of them moved their faces, causing the ROIs to be undetected. Quet et al. [59] claim that the maximum fluctuation for individuals is <10 bpm. In consequence, for this study, the difference between successive intervals to identify irregular beats was set to 10%. The procedure to clean the HR-RAW vector compared each interval *i* with the average value of the two adjacent intervals: previous and following. If the interval *i* had an average value either greater than or less than 10% of both intervals, it was eliminated because it was considered an irregular value. The procedure was repeated for the new interval *i*.

This procedure created a new dataset called HR-CLEAN. Figure 5 shows examples of the data removed from the HR-RAW data.

## 4. Results

A total of 56 first-year undergraduate students completed their profile (demographic data). These are the SXE/SXR baseline anxiety scales (pre-test). The students were video recorded during the math class. In addition, all of them completed the SXE scale during the exam. They were video recorded while doing this. The results of the 112 STAI assessments are as follows:59% of the students were prone to high levels of anxiety.33% of the students presented high anxiety as a result of an academic activity (class or exam).24% of the students were prone to high levels of anxiety but not necessarily as a result of academic activities.

From these observations, it can be seen that 59% of the students attending math classes are prone to high anxiety. From the observations gathered:33% of the students presented high anxiety as a result of the math class.34% of the students presented high anxiety during the exam.32% of the students presented high anxiety after the exam.

The sample used for feature analysis was the 56 students who completed their profile and SXE/SXR scales. From this sample, 11 observations were discarded for the following reasons. First, the participation of the students was voluntary, and although they had agreed to participate in the experiment, at some point in the experiment, they decided not to continue because they did not feel comfortable being recorded. Second, the camera position was fixed, and the participants were instructed to sit in front of the camera and always try to look straight ahead. However, some students had involuntary movements, and their faces went out of focus. Hence, 45 SXE observations were useful and their videos were processed using rPPG to calculate HR values. These values, in combination with the data from the students’ profile (age, gender, private or public school, sports participation, extra-curricular activities, family issues, and failed tests), were used to build the feature vector of each dataset. In addition, these features were associated with their label class of 0 (no anxiety observed) or 1 (anxiety observed). The cutoff scores of the SXE scale were set as a score of more than 43.01 for males and 43.69 for females. Scores higher or equal to these values were considered instances of high academic stress [60].

Five datasets were proposed and tested using four classifiers (KNN, SVM, RF, and J48) to find the dataset and classifier that provide the highest accuracy to identify academic stress due to anxiety reactions. Each classifier was tested with 10-fold cross-validation, with the parameters of each classifier set as described in Section 2.6. Table 3 and Table 4 and Figure 6, Figure 7, Figure 8, Figure 9 and Figure 10 show the confusion matrix, accuracy, sensitivity, specificity, Kappa, and AUC values of each dataset obtained by the different classifiers. The results are summarized below:Dataset 1—Demographic features. This dataset showed 83.71% to 86.26% accuracy for the four classifiers, with sensitivity in the range of 0.82 to 0.87, specificity of 0.82 to 0.85, Kappa values of 0.67 to 0.72, and AUC values of 0.84 to 0.95.Dataset 2—HR-RAW data. It was one the datasets with the worst results; accuracy values in a range of 50.28% to 54.10%, with sensitivity less than 0.58, specificity of 0.50 to 0.53, Kappa values less than 0.07, and AUC values of 0.49 to 0.57.Dataset 3—Demographic features in combination with HR-RAW data. This dataset showed 88.66% to 90.79% accuracy for the four classifiers, with sensitivity in the range of 0.86 to 0.98, specificity of 0.85 to 0.91, Kappa values of 0.77 to 0.81, and AUC values of 0.87 to 0.95.Dataset 4—HR-CLEAN data. This was one of the datasets with the worst results; accuracy values in the range of 54.39% to 60.00%, with sensitivity less than 0.62, specificity of 0.52 to 0.60, Kappa values less than 0.20, and AUC values of 0.55 to 0.61.Dataset 5—Demographic features in combination with HR-CLEAN data. It was the dataset with the best results. This dataset showed 94.47% to 96.45% accuracy, with sensitivity higher than 0.95, specificity of 0.90 to 0.96, Kappa values greater than 0.88, and AUC values of 0.94 to 0.98.

As can be seen in Figure 6, datasets 1, 3, and 5 obtained an accuracy greater than 80%. Datasets 2 and 4 obtained an accuracy close to 50%. In the case of sensitivity and specificity, Figure 7 and Figure 8 show that datasets 1, 3, and 5 obtained values greater than 0.80. Datasets 2 and 4 obtained values less than 0.60. In addition, Figure 10 demonstrates that the AUC of datasets 1, 3 and 5 for most classifiers was greater than 0.90. This means there is a 90% or more chance that any one of the classifiers will be able to distinguish between both classes. On the other hand, the AUC for datasets 2 and 4 indicates that there is a less than 60% chance that the classifiers will be able to distinguish between both classes. Finally, dataset 5 obtained the highest Kappa values of the four classifiers.

## 5. Discussion

As analyzed and described in Section 1, stress is a common condition in society. Since academic stress cannot be measured directly, other mechanisms are used to correlate it as a result either of behaviors or physiological signs. According to Rodríguez-Arce et al. [35], heart rate, blood oxygenation, and/or galvanic skin response are metrics that are strongly correlated to academic stress.

From the analysis in Section 1, it could be concluded that most of the methods for measuring academic stress as a body response are either used in a lab-controlled environment [24] or present a particular bias in measuring stress [25,35]. Hence, research that proposes an innovative method for analyzing academic stress makes an important contribution to the literature. Academic stress is derived from academic activities, which are considered the stressor. Anxiety state is considered a clinical consequence of academic stress. The proposed method is based on body responses observed in video recordings, rPPG, student demographic data, and machine learning classification algorithms.

The fundamental hypothesis of this study lies in the fact that academic stress can be detected in video recordings. As a consequence, a non-contact method that uses rPPG for identifying states of academic stress and its associated anxiety features is presented. This method was tested with 56 undergraduate students. No special conditions or set-up were required. Participants were video recorded during two experimental sessions. HR values were calculated from video recordings. With these values and the students’ profiles, a classification process was run for each student’s data. As described in Section 4, four machine learning classifiers were used and compared for performance: SVM, KNN, J48, and RF, using the Weka Machine Learning toolset. For each classification method, five datasets were tested.

In general, all performance indicators behave better for Datasets 1, 3, and 5. As presented in Section 4, Datasets 2 and 4 (HR values) obtain low-performance metrics for any of the classifiers in comparison with the metrics of Dataset 1 (students’ demographic data). However, when Dataset 1 is combined with the data from Datasets 2 and 4, the performance indicators are improved, as demonstrated by the results of Datasets 3 and 5. Nevertheless, when Kappa values are analyzed for Datasets 3 and 5, the latter obtains values greater than 0.80. This means that there is a perfect agreement between the anxiety values provided by the STAI scales and the labels calculated by the classifier. In the case of Dataset 1, although its performance indicators (such as accuracy, sensitivity, and specificity) are high, the Kappa values for any of the classifiers are near 0.60. This signifies a moderate agreement between the results of the STAI scales and the results of the classifiers. Regarding the AUC values, Datasets 1 and 3 obtain values in the range of 0.8 to 0.9 for most of the classifiers. According to Mandrekar [61], these values indicate an excellent predictor for identifying subjects with and without academic anxiety in comparison with Datasets 2 and 4. Nevertheless, Dataset 5 obtains AUC values greater than 0.90 for any of the classifiers. This is considered an outstanding predictor—hence the recommendation for using this dataset for academic stress classification.

When Kappa values are closely analyzed for Datasets 2 and 4, where the values are in the range of 0.0 to 0.20, one observes a minimal agreement between the anxiety values provided by the STAI scales and the labels calculated by the classifier. In such cases, the use of neither dataset is recommended for any of the classifiers.

These results can be contrasted with those of previous studies which used contact methods. For example, Subhani et al. [62] used a machine learning framework involving electroencephalogram (EEG) signal analysis of stressed participants. They reached 94.6% accuracy for two-level identification of stress. Gjoreski et al. [63] monitored stress with a wrist device. They showed that their method detects stress events with a precision of 95%. Rodríguez-Arce et al. [35] reached 95% accuracy by using three signals and six features. Other methods were less accurate. Sano et al. [64] used a wrist sensor and a mobile phone, and their accuracy was over 75% in a binary classification. Castaldo et al. [65] studied mental stress in oral academic examinations via ultra-short-term HRV analysis. Their results were 79% accurate. The main difference between previous studies and this research is that contact sensors are required for reaching those levels of accuracy. The method proposed here uses non-contact sensors and a simple webcam, which, combined with the students’ demographic data, provided a slightly better or similar level of accuracy than had been reached previously. Moreover, despite the fact that previous studies obtain similar performance indicators, the proposed rPPG-based system overcomes technical issues such as the size of the sample used, signal acquisition being highly sensitive to noise and artifacts causing miscalculation measures, manual cleaning of the datasets, and the method not being biased toward a commercial device.

### Academic Implications

Stress measurement, either with current methods or with psychology, requires considerable time for detailed analysis and can be a complex process. However, other implications have been observed, such as costs to public finance, the associated implications for public schools, and the academic performance of students.

Based on the initial experimentation, it has been demonstrated that it is possible to provide accurate academic stress information by using the STAI results. The SXR scale provides information on how a student could develop any level of academic anxiety. The SXE scale, on the other hand, provides information on the level of academic anxiety that a student experiences while doing a specific academic activity.

The results show that up to 70% of first-year undergraduate students are prone to high levels of anxiety (SXR). This finding aligns with Kitsantas’ research [11]. Engineering students tend to develop mid to high anxiety levels during the first year of university, but further studies should be conducted. In addition, this information provides insights into the potential for students to develop stress from any academic activity. Future studies should be focused on finding which activities might lead students to develop such levels of academic stress.

When dealing with stress associated with academic environments, the effects are not always immediate. Therefore, proper attention is required in order to avoid academic disenrollment. The associated cost of public university disenrollment is high. Our results show that classes, exams, and/or assignments are likely the cause of the high levels of anxiety observed in 59% of first year undergraduate students. Ros-González showed that 50% of disenrollment in first-year undergraduate students is associated with academic stress [12]. The proposed methodology is 96% accurate. It is therefore possible to identify stress in students based on anxiety reactions.

It is important to mention that our results demonstrate that on average, 47% of students in this experiment show high anxiety when taking a mathematics exam. This result means that at least one-half of the study group experiences academic stress as a result of exams. The main implication of our study is that by using the proposed method, it is possible to identify situations in which students may experience academic stress, such as classes, presentations, or exams. Eventually, the system might provide feedback to professors. In this way, methods could be devised to help students in near real time.

The results claim that the proposed classifiers (J48, random forest, and KNN) provide an accuracy of 96% with Kappa values of 0.92. The implementation of such classifiers in an academic stress detection tool in the classroom or during virtual remote sessions is achievable and might provide near real-time information. This could be used for a student feedback system to help manage their stress and anxiety levels.

The current methodology can be summarized in the following steps:It requires a demographic student profile.It requires video recordings of students’ faces while doing an academic task.The rPPG technique is applied to each video recording in order to obtain HR values.Pre-processing of the HR value signal is needed to eliminate noise.Stress levels might be obtained from the students’ data and HR-CLEAN values.

Finally, the current scope is limited to first-year undergraduate students in engineering school, but the methodology can be easily replicated in other schools. Moving forward, the work can be extended to any year, any school, and potentially to virtual environments. The rPPG’s algorithm might be enhanced and fully automated to support multi-facial recognition in a classroom.

## 6. Conclusions

Identifying academic stress in school is challenging and, as far as the authors know, current systems have not been applied explicitly to academic environments. As a consequence, this study focuses on the anxiety symptoms caused by academic stress in school environments as one of the clinical consequences.

In this way, this study proposes a non-contact method based on the use of rPPG to identify academic stress levels derived from academic activities in school environments. Fifty-six undergraduate students participated in an experimental protocol. HR values were calculated from video recordings using rPPG. Five datasets and four machine learning classifiers were run to identify which one provides the best performance metrics to identify subjects with and without academic anxiety. The results claim that the proposed methodology provides 96% accuracy by using KNN, J48, and random forest classifiers from the HR-CLEAN dataset in combination with seven features of students’ profiles. Moreover, our performance metrics show better results compared to the contact methods. As a consequence, the initial hypothesis is accepted.

The contributions of this study are the following: (a) an experimental methodology for identifying academic stress using rPPG in academic environments, (b) the proposed method does not require special conditions or a complex set-up, (c) academic stress assessments can be carried out in typical classroom conditions, (d) the proposed system overcomes the technical issues found in previous works, (e) the proposed method is not biased toward a commercial device, and (f) the proposed method might be helpful in making prompt decisions in terms of academic stress in undergraduate students.

Finally, with the current results, it may be possible to implement an embedded platform (such as Raspberry Pi) to obtain students’ profiles and facial video recordings in order to calculate the students’ stress levels in academic environments in an easy and low-cost way.

## Figures and Tables

**Figure 1 sensors-22-03780-f001:**
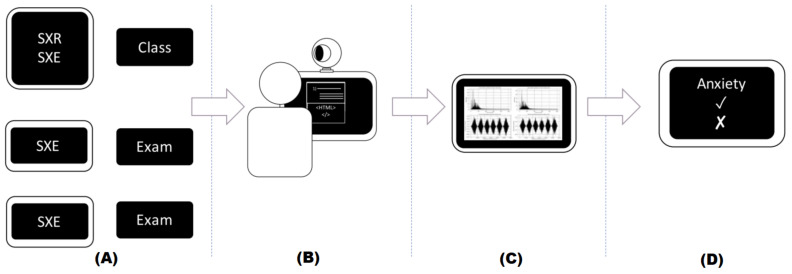
Experimental methodology for identifying academic stress using rPPG and demographic data. (**A**) Define academic stress protocol, (**B**) STAI, student’s profile and video recording in the academic environment, (**C**) Calculate HR signal from video recordings using rPPG, and (**D**) Classification of anxiety level using HR values.

**Figure 2 sensors-22-03780-f002:**
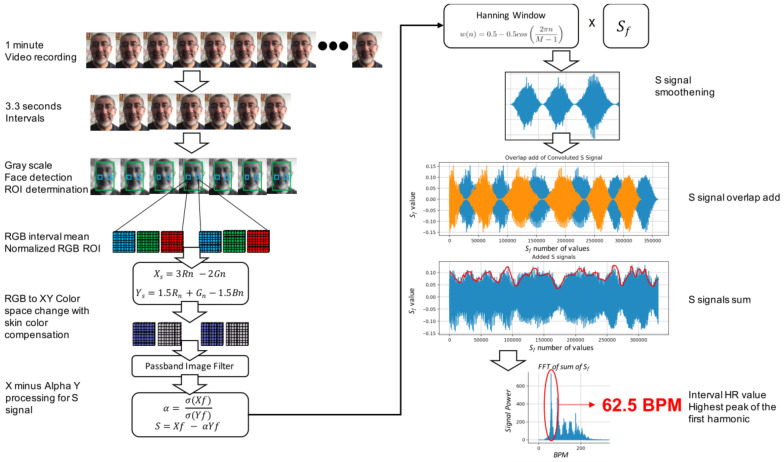
Chroma-based rPPG was used to calculate the HR values using ROI extracted from video recordings of students’ faces.

**Figure 3 sensors-22-03780-f003:**
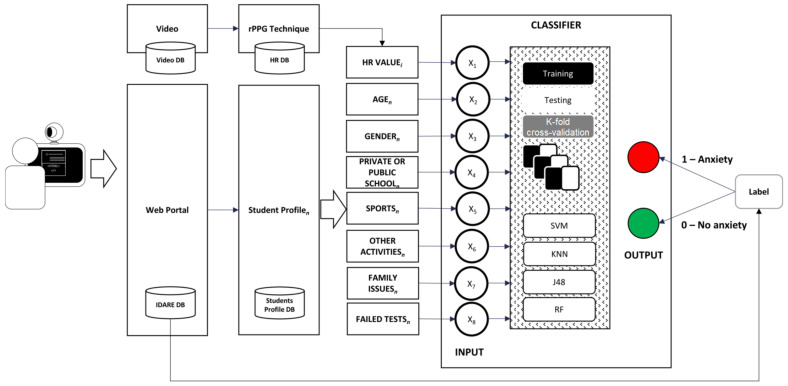
Classification phase uses eight features that are labeled to academic stress levels.

**Figure 4 sensors-22-03780-f004:**
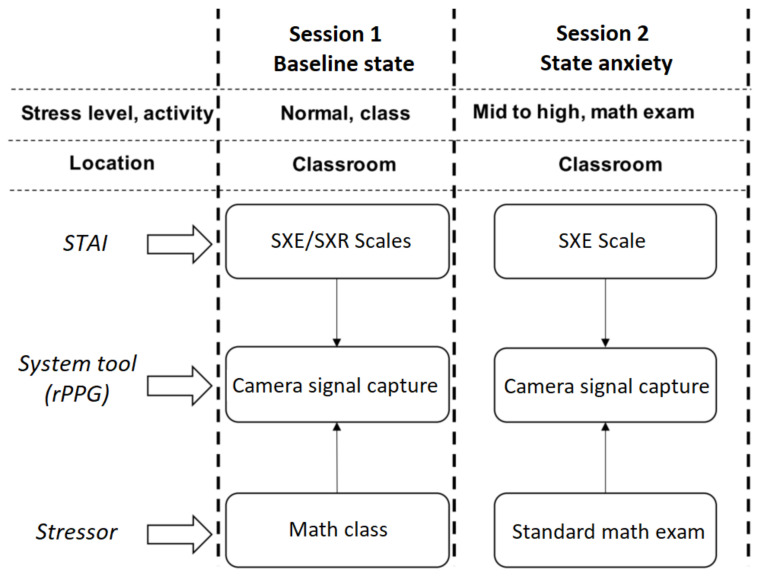
The academic stress protocol was designed to detect and assess anxiety during a class and measure changes during a written exam.

**Figure 5 sensors-22-03780-f005:**
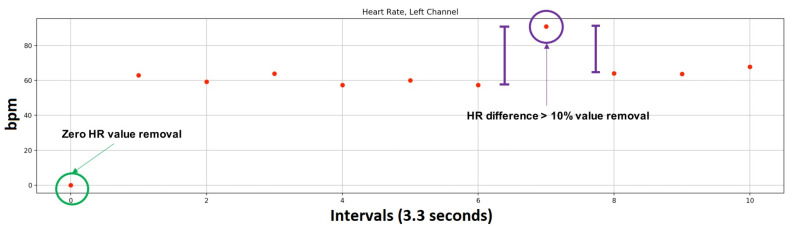
HR values from rPPG required noise removal by eliminating non-sense HR values from the HR-RAW data, such as zero and adjacent values with changes greater than 10%.

**Figure 6 sensors-22-03780-f006:**
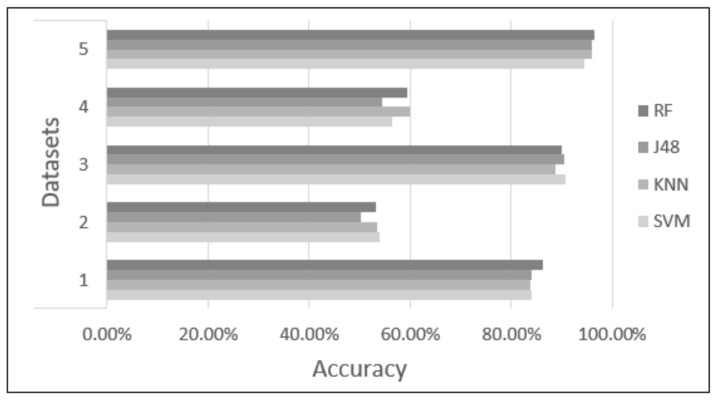
Difference in accuracy among the different classification methods.

**Figure 7 sensors-22-03780-f007:**
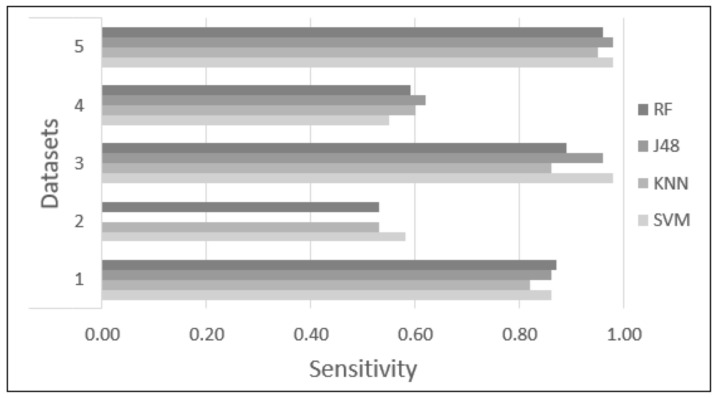
Difference in sensitivity among the different classification methods.

**Figure 8 sensors-22-03780-f008:**
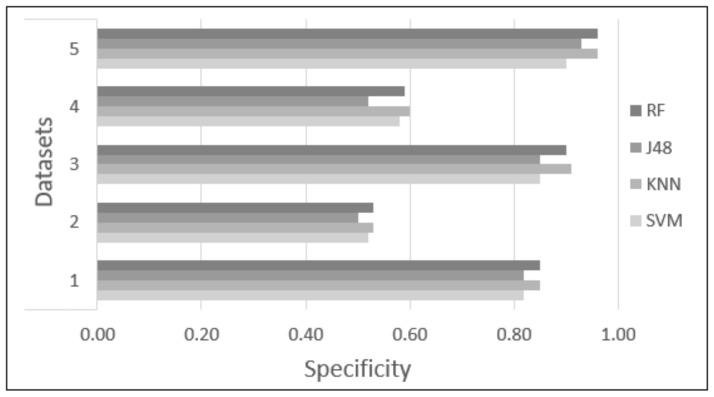
Difference in specificity among the different classification methods.

**Figure 9 sensors-22-03780-f009:**
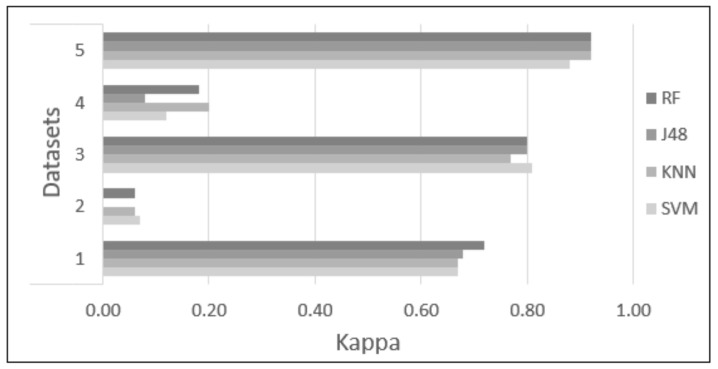
Difference in Kappa values among the different classification methods.

**Figure 10 sensors-22-03780-f010:**
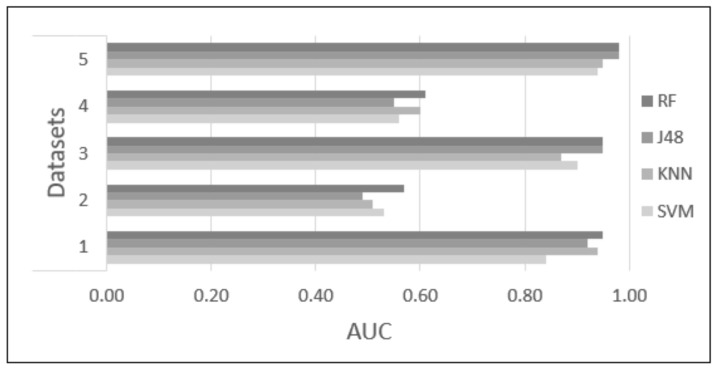
Difference in AUC values among the different classification methods.

**Table 1 sensors-22-03780-t001:** Video-based stress detection systems.

Type	Characterization
A	Webcam-based imaging detects small head movements and correlates them with biometrics such as pulse rate or respiration [37,38].
B	High-fidelity webcams with infrared and high-speed frame detection implement the PPG algorithm and correlate data with heart rate and heart rate variability [39,40].
C	Video frames, facial recognition and color bit analysis to reproduce a similar finger PPG LED/LRD (Light Emission Diode/Light Responsive Diode) optical reflection data using bit-pattern analysis on a region of the subject’s face [41,42,43,44].

**Table 2 sensors-22-03780-t002:** Cohen’s Kappa coefficient interpretation for classifiers accuracy (adapted from [57,58]).

Kappa Result	Interpretation	Percentage of Data Correct
<0	None agreement	0–4%
0.01–0.20	Poor agreement	4–15%
0.21– 0.40	Fair agreement	15–35%
0.41–0.60	Moderate agreement	35–63%
0.61–0.80	Substantial agreement	64–81%
0.81–1.00	Almost perfect agreement	82–100%

**Table 3 sensors-22-03780-t003:** Values of different measures for different classification methods.

Classifier	Dataset	Accuracy	Sensitivity	Specificity	Kappa	AUC
SVM	1	0.84	0.86	0.82	0.67	0.84
	2	0.54	0.58	0.52	0.07	0.53
	3	0.91	0.98	0.85	0.81	0.90
	4	0.56	0.55	0.58	0.12	0.56
	5	0.94	0.98	0.90	0.88	0.94
KNN	1	0.84	0.82	0.85	0.67	0.94
	2	0.53	0.53	0.53	0.06	0.51
	3	0.89	0.86	0.91	0.77	0.87
	4	0.60	0.60	0.60	0.20	0.60
	5	0.96	0.95	0.96	0.92	0.95
J48	1	0.84	0.86	0.82	0.68	0.92
	2	0.50	0.00	0.50	0.00	0.49
	3	0.90	0.96	0.85	0.80	0.95
	4	0.54	0.62	0.52	0.08	0.55
	5	0.96	0.98	0.93	0.92	0.98
RF	1	0.86	0.87	0.85	0.72	0.95
	2	0.53	0.53	0.53	0.06	0.57
	3	0.90	0.89	0.90	0.80	0.95
	4	0.59	0.59	0.59	0.18	0.61
	5	0.96	0.96	0.96	0.92	0.98

**Table 4 sensors-22-03780-t004:** Confusion matrix where the True Positive (TP), True Negative (TN), False Positive (FP), and False Negative (FN) values can be viewed clearly.

Classifier	Dataset	Confusion Matrix		Classifier	Dataset	Confusion Matrix	
SVM	1	283 (TP) 45 (FP)	68 (FN) 310 (TN)	J48	1	283 44	68 311
	2	91 64	260 291		2	0 0	351 355
	3	289 3	62 352		3	295 12	56 243
	4	234 191	117 164		4	73 44	278 311
	5	312 0	39 355		5	328 5	23 350
KNN	1	302 66	49 289	RF	1	298 44	53 311
	2	186 164	165 191		2	186 164	165 191
	3	322 51	29 304		3	318 37	33 318
	4	207 138	144 217		4	203 138	148 217
	5	338 15	13 340		5	338 12	13 343

## Data Availability

Not applicable.

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
