# Peer review of "Towards a Non-Contact Method for Identifying Stress Using Remote Photoplethysmography in Academic Environments"

_sensors, 2022, doi:10.3390/s22103780_

Round 1
Reviewer 1 Report
"Towards a non-contact method to identify stress using the remote photoplethysmography (rPPG) technique in academic environments" is an interesting article. The aim was to analyze the stress derived from academic activities considered as the stressor in order to propose a non-invasive method to identify states of stress in academic environments using the remote photoplethysmography technique in combination with demographic data of students. The results shown that the proposed methodology provides 96% accuracy using KNN, J48 and Random Forest classifiers from HR CLEAN data set (HR signal from rPPG) and 7 student’s profile to classify the presence or absence of stress derived from academic activities.
There are issues in the article that need to be addressed.
Abstract
Line 11, add the meaning of KNN
Introduction
Lines 37 to 44. The concepts of stress and anxiety are described in a confusing way. It is suggested to clearly explain these concepts. In addition, to differentiate anxiety as a symptom, the anxiety reaction, generalized anxiety disorder and the differences or similarities with stress.
Line 48. It must be substantiated why an instrument is used to measure anxiety if what is intended to be studied is stress, and why a specific instrument was not used to measure stress or academic stress.
Materials and Methods:
On line 190. ¿Is it possible to obtain the Heart Rate Variability in the frequency spectrum? It is known that, in addition to HR, it is important to evaluate HRV, which provides additional or even more important information on the sympathetic and parasympathetic tone of the heart and is modified in states of anxiety and stress even when HR is not modified.
Results:
Line 324 mentions that the videos of 11 subjects were eliminated, what were the specific reasons for said elimination; it is interesting to know this information to determine the feasibility of the proposed technique.
Information on the concept of accuracy and Cohen's Kappa coefficient, etc. expressed in lines 352 to 362 should be placed in the Materials and Methods section in addition to the statistical analysis used (including table 5). Su suggests defining and calculating positive and negative predictive values, sensitivity, specificity, and precision, and even representing the results by means of ROC curves. All of the above will give greater clarity to the results.
Discussion:
Throughout the article, in its title and especially in the discussion, the term stress is used in a confusing and inconsistent manner and is mixed with the term anxiety. Throughout the article and especially in the discussion, these concepts must be clarified, analyzed and discussed.
In most of the subsection “4.1. Technical analysis” there is no reference and it is a matter of summarizing the results, which is redundant. At the end (lines 418 to 427) some studies are mentioned, it is suggested, in addition to mentioning the results of said studies, to make a comparison and discussion of them in relation to those obtained by the authors.
References
It is suggested to review this section; there are some inconsistencies in the way of citing, for example, the name of the Journal does not have the correct font. Pages are cited differently)
Author Response
Please see the attachment.
The document has been reviewed by a native professor and he helped us make the necessary corrections.

Reviewer 2 Report
The manuscript introduces a remote PPG (rPPG) method to identify stress. The authors claim that the proposed rPPG method can achieve no less accuracy than that achieved by the contact PPG method. HR from rPPG is already reported method. The innovation is not enough for this manuscript. The author should highlight what's new they do that introduces an improvement. And the data should be separated as training and test data sets. I may suggest a major.
(1) Grammar mistake:
Kitsantas and Cols [2] research claims... (page 1 line 25)
(2) Why the absolute difference of 10% is set when conducting HR CLEAN?
(3) "if the absolute difference between interval i and i+1 was greater than 10%, then the value was deleted"
How about i+2 which also can have a >10% difference from i+1?
(4) How accuracy is defined mathematically in Table 2 - Table 4?
(5) To validate a model, do the authors separate the data as training and test data?
Author Response

(The authors gave the same response as above.)

Author Response

(The authors gave the same response as above.)

Round 2
Reviewer 2 Report
Thanks for the detailed revision, I feel the revised version is suitable for publication.
Reviewer 3 Report
The authors did a wonderful job in addressing my previous comments. I only have two additional minor comments:
- In Figure 3, it should be "k-fold cross-validation" instead of "cross-fold validation".
- The authors are suggested to trim the Conclusions -- it should not include lengthy details about hypothesis or specifics of method. Instead, it should briefly cover 1) the objectives of this study 2) what it did 3) description of results at high-level 4) what we learned from this study. Part 1-3 may have 1-2 sentences for each, and Part 4 may have a bit more than that. Conclusions is meant to be short, succinct, and to the point.
Thank you.
